# Pathogenesis, Intervention, and Current Status of Drug Development for Sarcopenia: A Review

**DOI:** 10.3390/biomedicines11061635

**Published:** 2023-06-04

**Authors:** Jung Yoon Jang, Donghwan Kim, Nam Deuk Kim

**Affiliations:** 1Department of Pharmacy, College of Pharmacy, Research Institute for Drug Development, Pusan National University, Busan 46241, Republic of Korea; jungyoon486@pusan.ac.kr; 2Functional Food Materials Research Group, Korea Food Research Institute, Wanju-gun 55365, Jeollabuk-do, Republic of Korea; kimd@kfri.re.kr

**Keywords:** sarcopenia, muscle strength, muscle atrophy, frailty, aging, senescence, skeletal muscle mass loss, intervention, clinical trials, treatment

## Abstract

Sarcopenia refers to the loss of muscle strength and mass in older individuals and is a major determinant of fall risk and impaired ability to perform activities of daily living, often leading to disability, loss of independence, and death. Owing to its impact on morbidity, mortality, and healthcare expenditure, sarcopenia in the elderly has become a major focus of research and public policy debates worldwide. Despite its clinical importance, sarcopenia remains under-recognized and poorly managed in routine clinical practice, partly owing to the lack of available diagnostic testing and uniform diagnostic criteria. Since the World Health Organization and the United States assigned a disease code for sarcopenia in 2016, countries worldwide have assigned their own disease codes for sarcopenia. However, there are currently no approved pharmacological agents for the treatment of sarcopenia; therefore, interventions for sarcopenia primarily focus on physical therapy for muscle strengthening and gait training as well as adequate protein intake. In this review, we aimed to examine the latest information on the epidemiology, molecular mechanisms, interventions, and possible treatments with new drugs for sarcopenia.

## 1. Introduction

The term sarcopenia (Greek, “*sarx*” meaning “flesh” and “*penia*” meaning “loss”) was first proposed by Rosenberg in 1997 and refers to the reduction of both muscular mass and function during the natural process of aging [1]. Decreased muscle strength negatively affects steady walking and contributes to a high incidence of falls among elderly individuals. Sarcopenia is strongly linked to self-reported physical disability in both men and women, regardless of ethnicity, age, morbidity, obesity, income, or health behaviors [2]. Reduced muscle strength with age leads to reduced functional ability and is a significant cause of disability, mortality, and other negative health outcomes [3]. As the number of elderly people continues to increase, there is a growing demand for healthcare resources to address sarcopenia-related morbidities.

Initially, the definition of sarcopenia only considered the loss of muscle mass and not muscle strength or physical impairment as part of the disease process [3]. However, in 2010, the European Working Group on Sarcopenia in Older People (EWGSOP) recognized sarcopenia as a syndrome characterized by the progressive and widespread loss of skeletal muscle mass and strength, with an increased risk of negative outcomes such as physical disability, poor quality of life, and mortality [4,5,6]. They acknowledged that loss of muscle mass and strength and reduction in physical performance are important characteristics of sarcopenia. Sarcopenia can occur in sarcopenic obesity in which both muscle mass loss and obesity occur simultaneously, and osteosarcoma visceral obesity can result in bone and muscle loss and accumulation of fat in the abdomen [7]. In 2018, the EWGSOP revised its diagnostic criteria to consider low muscle strength (LMS) as the primary parameter for diagnosing sarcopenia. Muscle strength is currently considered the most reliable measure of muscle function according to the revised guidelines (EWGSOP2) [8]. The updated guidelines also recognize muscle strength as a more effective predictor of adverse outcomes than muscle mass [9,10,11,12].

Timely identification and treatment can help prevent the negative outcomes associated with sarcopenia. Despite being recognized as a significant clinical issue, sarcopenia has only recently been officially categorized as a disease. In 2016, sarcopenia was added to the International Classification of Diseases, Tenth Revision, Clinical Modification (ICD-10-CM) by the United States, a global disease classification system published by the World Health Organization, and given a specific code (M62.84) [13]. The eighth revision of the Korean Classification of Diseases (KCD) in 2021 also included sarcopenia as a disease and assigned it an ICD-10-CM code (M62.5) [14].

## 2. Epidemiology and Pathophysiology

Sarcopenia is prevalent in older individuals, with rates ranging from 11% to 50% in those over 80 years of age and up to 29% in community healthcare settings [15]. This condition is characterized by an imbalance between anabolic and catabolic processes, leading to a decline in the size and number of type II muscle fibers and intramuscular and intermuscular fat infiltration. Additionally, the number and function of satellite cells, responsible for repairing damaged muscle fibers, decrease [16,17,18]. Several factors, including neuromuscular junction dysfunction, decreased number of motor units [19], inflammation [20], insulin resistance [21], mitochondrial dysfunctions [22,23], and oxidative stress [24], contribute to muscle loss in sarcopenia. Moreover, the denervation of single muscle fibers can cause a substantial reduction in type II fibers, which are replaced by type I fibers and fat tissue. These changes in muscle structure and function can result in adverse outcomes such as physical disability and poor quality of life. Early recognition and intervention can mitigate these negative consequences.

## 3. Risk Factors for Primary Sarcopenia

Most people believe that sarcopenia is an inevitable aspect of the aging process; however, its severity can vary significantly and is influenced by particular risk factors. Therefore, classification of muscle atrophy as primary or secondary sarcopenia may be useful in clinical practice [8]. The following sections summarize the risk factors for primary sarcopenia:

### 3.1. Lack of Exercise 

Insufficient exercise is considered a primary risk factor for sarcopenia [25]. Around the age of 50 years, a gradual reduction in the number of muscle fibers begins to occur [26]. The reduction in muscle fiber and strength is more evident in sedentary individuals than in those who engage in physical activity. However, even professional athletes, such as weightlifters and motorcycle runners, show a gradual decrease in strength and speed with age, albeit at a slower rate [26].

### 3.2. Imbalance in Hormones and Cytokines

Age-associated reductions in the levels of hormone, such as growth hormone, testosterone, thyroid hormone, and insulin-like growth factor, can result in the loss of muscle mass and strength. In many cases, severe muscle loss arises due to a combination of reduced hormonal anabolic signals and the promotion of catabolic signals influenced by pro-inflammatory cytokines, such as interleukin-6 (IL-6) and tumor necrosis factor alpha (TNF-α) [27]. Research has revealed that both IL-6 and TNF-α levels are elevated in the skeletal muscles of older individuals [28,29].

### 3.3. Insufficient Protein Synthesis

The inability of the body to effectively synthesize protein, combined with an insufficient intake of calories and/or protein to maintain muscle mass, is frequently observed in sarcopenia. With aging, oxidized proteins within the skeletal muscle increase, leading to the accumulation of lipofuscin and cross-linked proteins that are not adequately cleared by the proteolysis system. Consequently, non-contractile and dysfunctional proteins accumulate in skeletal muscles, contributing to the significant decline in muscle strength observed in sarcopenia [30].

### 3.4. Dysfunction of Motor Units

Another age-related issue is the decline in motor neurons, responsible for transmitting signals from the brain to the muscles to initiate movement. Satellite cells are small, single-nucleus cells that line muscle fibers and are typically activated during exercise or injury. These signals prompt the differentiation of satellite cells, which then merge with muscle fibers, thereby aiding in the maintenance of muscle function. It has been proposed that sarcopenia is partially due to an inability to activate satellite cells [27].

### 3.5. Lifestyle of Individual

Evolutionary theories propose that genes regulating muscle mass and function are responsible for the inability of the body to maintain these traits with age [31]. According to this hypothesis, the genes that supported the high levels of muscle strength required for survival in the Late Paleolithic period are no longer necessary in the modern lifestyle, which is characterized by prolonged periods of sedentary behavior.

### 3.6. Physical Condition at Birth

Studies on the developmental origins of health and disease have revealed that environmental factors during early growth and development can have long-lasting effects on human health. Low birth weight, indicative of a suboptimal early environment, has been linked to low muscle mass and strength in adulthood [32,33]. Research has shown that a decrease in muscle fiber score is significantly associated with low birth weight, indicating that developmental factors affecting muscle physiology may account for the relationship between low birth weight and sarcopenia [34].

### 3.7. Nutritional Status

Malnutrition is a significant contributor to the development of sarcopenia, and a multidisciplinary approach should be implemented for its management, including nutritional screening and care plans similar to those used for cachexia [35,36]. Protein–energy malnutrition, along with other factors, is frequently associated with sarcopenia [37]. Maintaining a high protein intake above the Recommended Daily Allowance, in the range of 1.2–1.6 g/kg per day, has been suggested as a preventative measure for age-related sarcopenia [38].

## 4. Risk Factors for Secondary Sarcopenia

Sarcopenia is frequently associated with other medical conditions, and understanding the mechanisms that lead to muscle loss in secondary sarcopenia can offer valuable information regarding age-related sarcopenia [8]. The approach to managing secondary sarcopenia should prioritize the treatment of the primary underlying condition using the methods mentioned earlier to enhance skeletal muscle mass and strength.

### 4.1. Cachexia

Severe muscle wasting is often associated with severe systemic diseases such as cancer, cardiomyopathy, and end-stage renal disease, known as cachexia [39]. Cachexia is defined as a metabolic syndrome associated with an underlying illness, characterized by the loss of muscle and fat mass [40], and is frequently linked to inflammation, insulin resistance, anorexia, and increased breakdown of muscle proteins [41,42]. Although sarcopenia is one of the elements of the proposed definition of cachexia, not all sarcopenic individuals are considered cachectic [40]. Based on this information, a consensus definition has been developed by the Special Interest Group on cachexia–anorexia in chronic wasting diseases of the European Society for Clinical Nutrition and Metabolism (ESPEN-SIG) to differentiate between cachexia and other conditions associated with sarcopenia [43].

### 4.2. Frailty

Frailty is a syndrome that affects older adults and is caused by a gradual decline in multiple physiological systems with age, resulting in a reduced capacity to handle stress and reduced homeostatic reserve. This syndrome is associated with an increased risk of negative health outcomes, such as falls, institutionalization, hospitalization, and mortality [44]. Frailty is defined by the presence of three or more of the following criteria: unintentional weight loss, weakness, exhaustion, slow gait speed, and low physical activity [44,45]. Frailty and sarcopenia are often associated, with most frail older adults experiencing sarcopenia, suggesting a shared underlying mechanism. Frailty extends beyond physical characteristics and may also include psychological and social dimensions, such as cognitive decline, lack of social support, and environmental factors [45].

### 4.3. Sarcopenic Obesity

Sarcopenic obesity (SO) occurs when a person has low lean body mass due to sarcopenia combined with high fat mass. This medical condition has been linked to a range of problems, including impaired functional capacity, metabolic issues, disability, and increased mortality rates [8,46]. The reported prevalence of SO varies widely, ranging from 2% to 21.7%, and may be due to factors such as a lack of awareness among healthcare providers, genetic differences, nutrition, and lifestyle factors [47]. Under certain conditions, such as malignancy and rheumatoid arthritis, individuals may experience a loss of lean body mass while preserving or even increasing fat mass [46]. Furthermore, low muscle mass combined with high fat mass is a typical aspect of the aging process. However, identifying SO in older people can be challenging because the reduction in muscle mass and strength due to aging may occur independently of body mass index.

For a long time, it was believed that the age-related decline in weight and muscle mass was the main cause of muscle weakness in the elderly [48]. However, recent research has suggested that changes in muscle composition also contribute to this weakness. In particular, marbling or fat infiltration into the muscles reduces muscle quality and function [49]. Studies examining the development of sarcopenic obesity have identified age-associated changes in muscle and fat composition. For example, older men experience an initial increase in the percentage of fat mass, which later stabilizes or decreases [49]. There is also a redistribution of fat in the body, with an increase in intramuscular and visceral fat and a decrease in subcutaneous fat [50,51]. These changes may play a role in the development of SO.

## 5. Pathogenesis

The activation, differentiation, and proliferation of myoblasts in skeletal muscles are crucial for the regeneration of muscle fibers with mechanical, chemical, or degenerative damage [52,53]. The differentiation of myoblasts into myotubes is vital for the development and regeneration of skeletal muscles because these monocytic myoblasts proliferate, differentiate, and eventually fuse with existing muscle fibers to form multinucleated myotubes and myofibers [54]. The process of muscle cell proliferation and differentiation occurs during birth and postnatal development [55]. Hence, promoting the proliferation and differentiation of myoblasts and inducing myotube hypertrophy could potentially benefit muscle regeneration and the preservation of muscle mass. 

The main mechanism for muscle regeneration is to activate serine/threonine kinase, which amplifies mammalian target of rapamycin (mTOR), ultimately leading to an increase in muscle protein synthesis [56]. This process is stimulated by anabolic factors such as insulin-like growth factor-1 (IGF-1), testosterone, and exercise. Testosterone also plays a role in repairing muscle cells by promoting myoblasts and inhibiting myostatin. However, with aging, the levels of testosterone, IGF-1, and exercise decrease, resulting in a decrease in muscle protein synthesis and an increase in muscle breakdown. There are several signals that contribute to muscle atrophy, including aging, cachexia, diabetes, disuse, fasting, chronic inflammation, malnutrition, motor neuron loss, uremia, and dexamethasone. These factors lead to the development of sarcopenia through various signaling pathways. However, this paper provides a brief summary without mentioning the detailed molecular mechanisms of each factor (Figure 1). Muscle wasting occurs due to increased protein breakdown, which is caused by activation of the ubiquitin–proteasome pathway [57]. The muscle-specific ligases muscle atrophy F-box (MAFbx; also known as atrogin-1) and muscle RING Finger-1 (MuRF-1) are expressed early in the muscle atrophy process and play crucial roles in muscle protein degradation such as myogenic differentiation 1 (MyoD) and myogenin [56,57,58,59,60,61]. Insulin resistance, elevated proinflammatory cytokines, genetic factors, and poor nutritional status (including insufficient energy and protein intake leading to weight loss and decreased levels of vitamin D) can further accelerate muscle breakdown [56,58]. 

## 6. Diagnosis of Sarcopenia

An accurate definition is crucial for the diagnosis of sarcopenia. In 2010, the EWGSOP proposed three criteria based on muscle mass, strength, and physical performance [6]. Low muscle mass was defined as a skeletal muscle mass (SMM) index of less than 8.90 kg/m^2^, handgrip strength, indicative of muscle strength, below 30 kg in men and 20 kg in women, and gait speed, indicative of physical performance, less than 0.8 m/s. To confirm the diagnosis of sarcopenia, the presence of low muscle mass and either LMS or low physical performance (LPP) is required. Sarcopenia is classified into pre-sarcopenia, sarcopenia, and severe sarcopenia, based on the presence of low muscle mass and functional impairment. However, in 2018, the EWGSOP2 revised its guidelines and regarded LMS as the primary parameter for sarcopenia diagnosis because it is considered the most reliable measure of muscle function [62]. The revised guidelines recognize that muscle strength is more effective than muscle mass in predicting adverse outcomes [9,10,11,12,63,64,65,66]. Sarcopenia is probable when LMS is detected and the diagnosis is confirmed by the presence of low muscle quantity or quality. However, using muscle quality, which involves micro- and macroscopic aspects of muscle architecture and composition, as the primary parameter for defining sarcopenia is challenging owing to technological limitations [67,68,69].

Severe sarcopenia is diagnosed when there is LMS, low muscle quantity or quality, or LPP. Patients who exhibit indications or symptoms of sarcopenia, such as muscle wasting, weight loss, falls, weakness, difficulty in rising from a chair, or slow gait speed, should undergo case-finding during clinical practice. To confirm the diagnosis, the EWGSOP2 recommends the use of either the SARC-F questionnaire, which consists of five items related to strength, assistance in walking, rising from a chair, climbing stairs, and falls [70,71,72], or the Ishii screening tool, which includes age, grip strength, and calf circumference as variables [73]. To evaluate skeletal muscle power, grip strength can be measured using Jamar or Smedley instruments for the arms [74] and the chair stand test, which involves performing five sit-to-stand repetitions in 30 s intervals, for the legs [75,76].

There are various methods for estimating muscle quantity or mass, such as dual energy X-ray absorptiometry (DXA) or bioelectrical impedance analysis (BIA); the results can be adjusted based on height or body mass index (BMI) [77,78]. Muscle quantity can be expressed in different ways, including total body SMM, appendicular skeletal muscle mass (ASM), or the muscle cross-sectional area of specific muscle groups using computed tomography (CT) or magnetic resonance imaging (MRI) [79,80,81,82]. Muscle mass and body size are correlated; therefore, it is important to adjust for body size. Although calf circumference is not a good measure of muscle mass, it predicts performance and survival in older people [83,84,85]. Physical performance, which involves both muscle and nerve function, can be measured using various tests such as gait speed, short physical performance battery (SPPB), timed up and go test (TUG), and 400 m walk test [6,86,87,88,89,90,91].

However, using Western criteria for diagnosing sarcopenia may not be appropriate for individuals residing in other continents, including Asia, who generally have a smaller body size, more adipose tissue, and a more physically active life style than Western people. Therefore, both the International Working Group on Sarcopenia (IWGS) in 2011 [92,93] and the Asian Working Group for Sarcopenia (AWGS) in 2014 worked on developing a consensus for sarcopenia diagnosis based on evidence from research studies conducted within certain regions. In 2014, the AWGS proposed an algorithm for sarcopenia diagnosis in Asians based on the EWGSOP guidelines but with clearly defined cutoff values for each diagnostic component [94]. These cutoff values included muscle mass measurements using DXA or BIA, handgrip strength, and usual gait speed. The AWGS revised its algorithm for sarcopenia diagnosis at a consensus meeting held in Hong Kong in May 2019 [95].

The AWGS 2019 maintained the previous definition of sarcopenia but revised the diagnostic algorithm and some criteria. For handgrip strength, the cutoffs are <28.0 kg for men and <18.0 kg for women. Physical performance can be assessed using the 6 m walk test (with a cutoff speed of ≤1.0 m/s), SPPB (with a cutoff score of ≤9), and the five-time chair stand test (with a cutoff time of ≤12 s) [6,94,96]. Additionally, calf circumference (men: <34 cm; women: <33 cm), SARC-F (with a score of ≥4 indicating sarcopenia), and SARC-Calf (with a score of ≥11 indicating sarcopenia) were used. The AWGS 2019 also introduced “possible sarcopenia” as a diagnosis when LMS is present with or without reduced physical performance. Low ASM plus LMS or LPP are considered to indicate sarcopenia, whereas low ASM, LMS, and LPP are considered to indicate severe sarcopenia. Finally, the original cutoffs for LMM in sarcopenia diagnosis were retained: <7.0 kg/m^2^ for men and <5.4 kg/m^2^ for women using DXA and <7.0 kg/m^2^ for men and <5.7 kg/m^2^ for women using BIA.

## 7. Histopathology

In its early stages, sarcopenia is marked by a decline in muscle size, followed by a deterioration in muscle quality over time. The decline in muscle tissue quality is caused by the replacement of muscle fibers with fat, increased fibrosis, metabolic changes, oxidative stress, and degeneration of neuromuscular junctions. As a result of these changes, there is a gradual loss of muscle function leading to frailty [27].

Studies have shown that sarcopenia mainly affects type II (fast twitch) muscle fibers, whereas type I (slow twitch) fibers are less affected [97]. The reduction in the size of type II fibers in sarcopenia can be up to 50%, which is moderate compared with the overall muscle mass reduction, suggesting that sarcopenia involves a decrease in the number and size of muscle fibers. Studies comparing muscle cross-sections of elderly and younger individuals have shown a significant reduction in both type I and type II fibers in the elderly [98]. Anatomical and electrophysiological studies also indicate a loss of motor neurons with age, suggesting that a chronic neuropathic process may contribute to a reduction in muscle mass [99,100]. Histological changes observed in sarcopenia may also be influenced by lifestyle, hormones, inflammatory cytokines, and genetic factors.

## 8. Intervention

Early detection and timely intervention are crucial for improving outcomes in patients with sarcopenia. Screening elderly patients for any physical function or impairment of daily activities during routine healthcare visits is recommended. Patients who have difficulties with activities of daily living should undergo specific testing for sarcopenia using the methods described earlier. It is also important to assess the living environment for fall hazards and take appropriate safety measures as part of the treatment plan.

### 8.1. Non-Pharmacologic Treatment

An inactive lifestyle is associated with a loss of muscle mass and strength. Therefore, exercise is considered a crucial component in the management of sarcopenia. Short-term resistance exercise enhances the capacity of the skeletal muscles to produce proteins [101]. Resistance training (RT) and strength training (ST) are effective interventions for sarcopenia prevention and treatment. RT has a positive effect on the neuromuscular system, hormone levels, and protein synthesis rates [102]. There is evidence that RT, aerobic exercise, balance training, and even walking are effective in preventing primary and secondary sarcopenia but also secondary sarcopenia [103]. A recent meta-analysis showed that a combination of dietary supplements and exercise may have some benefits in treating sarcopenia; however, the results varied across populations [104,105].

### 8.2. Pharmacological Therapies

Currently, no United States Food and Drug Administration (FDA)-approved drugs are available for the treatment of sarcopenia. The use of steroid hormones, such as dehydroepiandrosterone (DHEA), testosterone, and anabolic steroids, has been studied as a potential treatment for sarcopenia. While these agents have shown some positive effects on muscle strength and mass, their use is limited owing to adverse effects, such as an increased risk of prostate cancer in men, virilization in women, and an overall high risk of cardiovascular events [106,107].

Novel sarcopenia treatments are currently being developed and tested in clinical trials. One potential therapy is selective androgen receptor modulators (SARMs), which can selectively target androgen receptors and promote muscle growth. The mechanism of action of SARMs involves their ability to selectively activate androgen receptors in muscle and bone tissues while minimizing the activation of androgen receptors in other tissues, such as the prostate gland [108]. As of April 2023, several SARMs had been tested in clinical trials, including GTx-024 (also known as enobosarm, ostarine, or MK-2866), GSK2881078, RAD140, and S-23. These trials primarily focused on evaluating the safety and efficacy of SARMs in treating muscle-wasting conditions and improving physical function in older adults. We have summarized the clinical trial status of the representative SARMs GTx-024 and GSK2881078 (Table 1). 

In a Phase 2A clinical trial (NCT03359473, as of 10 April 2023), GSK2881078 was studied in conjunction with exercise in patients with chronic obstructive pulmonary disease (COPD) and impaired physical function [120]. The results showed that GSK2881078 increased leg strength in men but not in women. Lean body mass also increased; however, there were no improvements in the patient-reported outcomes. Some safety concerns were noted, including a reversible reduction in high-density lipoprotein cholesterol and transient elevation in hepatic transaminase levels. Nonetheless, GSK2881078 was well tolerated, and short-term treatment increased leg strength to a greater extent than physical training alone in men with COPD. SARMs hold promise for achieving gains in skeletal muscle mass and strength without the adverse effects associated with other anabolic steroids [120]. Although SARMs have shown promise in clinical trials, they have not yet been approved for use by the US FDA or other regulatory agencies. As with any new drug, more research is needed to fully understand the potential benefits and risks of SARMs and determine the appropriate dosages and treatment regimens for different patient populations.

Myostatin inhibitors are a class of drugs that target the protein myostatin, which is a negative regulator of muscle growth [121,122,123,124]. Myostatin inhibitors block the action of myostatin, thereby allowing for increased muscle growth and preventing muscle loss, especially in conditions such as sarcopenia. The mechanism of action of myostatin inhibitors against sarcopenia involves their ability to inhibit myostatin activity in muscle tissue. Myostatin normally binds to and activates a receptor on the surface of muscle cells, activin receptor type IIB (ActRIIB), which suppresses muscle growth and protein synthesis [125]. When myostatin inhibitors are administered, they bind to and neutralize myostatin, thereby preventing its binding to the ActRIIB receptor and suppressing muscle growth. This leads to increased protein synthesis and muscle growth, resulting in improved muscle mass and function. Furthermore, myostatin inhibitors have other beneficial effects on muscles, including increased satellite cell activation and differentiation, improved mitochondrial function, and decreased inflammation [121,122,123,124,125]. As of April 2023, several myostatin inhibitors have been tested in clinical trials, including apitegromab (SRK-015), bimagrumab (BYM338), domagrozumab (PF-06252616), landogrozumab (LY2495655), taldefgrobep alfa (BMS 986089), trevogrumab (REGN1033), and rAAV1.CMV.huFollistatin344. These trials primarily focused on evaluating the safety and efficacy of myostatin inhibitors in treating muscle-wasting conditions and improving physical function in older adults. We have summarized the clinical trial status of representative myostatin inhibitors (Table 2). However, myostatin inhibitors are still in the experimental stage of development and have not yet been approved for clinical use. Further research is needed to determine their long-term safety and efficacy and the optimal dosage and duration of treatment for sarcopenia.

Vitamin D is a fat-soluble vitamin that plays a vital role in bone metabolism, calcium absorption, and muscle function. Increasing evidence suggests that vitamin D deficiency is associated with an increased risk of sarcopenia, and vitamin D supplementation has been investigated as a potential treatment option for this condition [167,168]. The mechanism of action of vitamin D in sarcopenia involves its ability to modulate muscle function and protein synthesis. Vitamin D receptors are present in skeletal muscle tissue, and vitamin D stimulates protein synthesis and increases muscle mass and strength by promoting the activity of the mTOR signaling pathway, a key regulator of muscle growth [168]. Moreover, vitamin D improves muscle function by enhancing neuromuscular junction functions and reducing inflammation in muscle tissue. Vitamin D deficiency is associated with reduced muscle strength and an increased risk of falls and fractures in older adults. In addition to its effects on muscle function, vitamin D is also important for maintaining bone health. Vitamin D deficiency can lead to osteoporosis, which is another age-related condition associated with an increased risk of fractures. Overall, the mechanism of action of vitamin D in sarcopenia involves its ability to promote muscle function and protein synthesis, reduce inflammation, and prevent osteoporosis [169]. Additionally, previous studies have reported that in older adults with sarcopenia, consuming a whey protein-based nutritional diet rich in leucine and vitamin D improves muscle mass as well as physical function and function and reduces treatment intensity and cost [170]. As of April 2023, 23 clinical trials related to the treatment of sarcopenia using vitamin D have been registered, most of which have been completed, whereas some are currently recruiting participants. However, the optimal dose and duration of vitamin D supplementation for the treatment of sarcopenia remain under investigation, and further research is required to determine the long-term safety and efficacy of this approach [171,172].

Growth hormone (GH), produced by the pituitary gland, is involved in the regulation of growth and metabolism. GH has anabolic effects on muscle tissues, and its use has been investigated as a potential treatment for sarcopenia. The mechanism of action of GH against sarcopenia involves its ability to stimulate the production of IGF-1, which is a hormone that promotes muscle growth and protein synthesis [173]. GH stimulates the liver and other tissues to produce IGF-1, which acts on muscle cells to stimulate protein synthesis and increase muscle mass. In addition to its anabolic effects, GH improves muscle strength and endurance, possibly by increasing the number of motor units in the muscle tissue and enhancing neuromuscular function [174]. However, the use of GH as a treatment for sarcopenia is controversial and its long-term safety and efficacy are still being studied [175]. GH supplementation is associated with several side effects including fluid retention, joint pain, and an increased risk of diabetes and cardiovascular disease. As of April 2023, there have been two clinical trials related to the treatment of sarcopenia using GH, one of which has been completed, whereas the other is currently recruiting participants. The completed clinical trial [“Effects of an Oral GH Secretagogue (MK-677) on Body Composition and Functional Ability of Older Adults (MOT089; NCT00474279)”] evaluated the effects of MK-677, an orally active GH secretagogue, on body composition and functional ability of older adults. The study involved 65 healthy older adults aged 60–81 years who were randomized to receive either MK-677 or placebo for 12 months. The results of the study showed that the MK-677 treatment resulted in a significant increase in lean body mass and a decrease in fat mass compared to the placebo. Additionally, participants in the MK-677 group showed improvements in physical function and mobility, as measured by a battery of functional tests. However, the study also reported some adverse effects associated with MK-677 treatment, including transient increases in fasting glucose and insulin levels and mild edema and muscle pain [176]. Further research is required to fully understand the potential benefits and risks of using MK-677 to treat age-related changes in body composition and functional abilities. Therefore, GH therapy should only be considered in carefully selected patients under the supervision of healthcare professionals.

Several other compounds, including angiotensin-converting enzyme inhibitors and eicosapentaenoic acid, are being investigated [103,104]. To prevent sarcopenia via the treatment of cachexia, 11 compounds, including thalidomide, OHR/AVR118, celecoxib, VT-122, omega-3 supplements, and anabolic agents such as ghrelin and its analogs, MT-102, and ruxolitinib, have been studied [177]. Among them, MT-102, the first anabolic catabolic transforming agent, has been tested in a phase-II clinical study for treating cachexia in patients with late-stage cancer, and the results indicated a significant increase in body weight compared to placebo treatment [178]. In aged animal models, MT-102 has been shown to reverse sarcopenia, and further studies are currently underway to investigate its potential as a treatment for sarcopenia [179,180].

## 9. Exercise Mimetics

The most effective treatment for sarcopenia involves combining a consistent exercise program with essential amino acid supplementation. However, certain individuals, such as those with severe sarcopenia, severe frailty, hip fracture, congenital neuromuscular disorders, or in intensive care, are unable to engage in regular exercise and thus cannot benefit from its advantages. In such cases, pharmaceuticals known as “exercise mimetics” or “exercise in a pill” are the only available therapeutic strategies to partially replicate some of the benefits associated with exercise in these specific populations [125]. While no single pharmaceutical compound can fully replicate all the benefits of exercise, research has focused on exploring the benefits of activating specific signaling pathways associated with exercise. One of these molecular targets is the peroxisome proliferator-activated receptor delta (PPARδ or PPARβ), which plays a role in mediating the effects of exercise.

Overexpressing PPARδ and administering the PPARδ agonist GW501516, the expression of genes related to mitochondrial content such as uncoupling protein 3 (*Ucp3*), carnitine palmitoyltransferase 1b (*Cpt1b*), and pyruvate dehydrogenase kinase 4 *(Pdk4*) increased [181]. When combined with exercise, GW501516 also enhanced oxidative myofibers and running endurance in adult mice. In mouse models with muscular dystrophy, GW501516 improved skeletal muscle mass [182], although it did not have an effect on muscle mass in rats [183]. However, clinical trials in humans were discontinued due to a significant increase in cancer occurrence in multiple organs observed in preclinical studies using rats and mice treated with GW501516 [184].

Another example of exercise mimetics is the activator of adenosine monophosphate-activated protein kinase (AMPK) [125]. AMPK is an enzyme that plays a crucial role in regulating energy metabolism and maintaining cellular homeostasis. Activation of AMPK has positive effects on muscle metabolism, including increased glucose uptake, mitochondrial biogenesis, and muscle protein synthesis. Some compounds, such as 5-aminoimidazole-4-carboxamide ribonucleotide (AICAR), have been shown to activate AMPK and improve muscle function [185]. In addition to PPARδ agonist and AICAR, there have been numerous studies on exercise mimetics utilizing peroxisome proliferator-activated receptor γ coactivator-1α (PGC-1α) modulators, nuclear factor erythroid 2–related factor 2 (Nrf2) modulators, irisin, and others [186,187,188]. However, currently, there are no exercise mimetics that are used in clinical practice, and further research is needed to clarify the molecular mechanisms and other aspects. Additionally, clear clinical studies are required to investigate the preventive efficacy of exercise mimetics in sarcopenia.

## 10. Herbal Supplements and Nutrition

Herbal supplements are of significant interest owing to their potential to promote muscular mass and health in patients with sarcopenia. Recent reviews identified numerous herbal compounds with effects on skeletal muscles [189,190]. Several of these compounds have shown mild effects on the skeletal muscles in human studies. For instance, curcumin from *Curcuma longa*, alkaloids and steroidal lactones from *Withania somnifera* (Solanaceae), catechins from *Camellia sinensis*, proanthocyanidin of grape seeds, and gingerols and shogaols from *Zingiber officinale* were found to have positive effects on skeletal muscle in human studies [189].

Various phytochemicals in rosemary have been studied for their effects on skeletal muscle cells, and several results have been reported. For instance, rosmarinic acid increases glucose uptake in skeletal muscle cells and activates 5′-adenosine monophosphate-activated protein kinase (AMPK) [191]. Carnosol attenuates C2C12 myotube atrophy and reduces insulin resistance in muscle cells and adipocytes [192,193]. Ursolic acid ameliorated indoxyl sulfate-induced mitochondrial biogenesis disorders in C2C12 cells [194], and together with leucine, it potentiated the differentiation of C2C12 murine myoblasts through the mTOR signaling pathway [195]. An ethanol extract of loquat (*Eriobotrya japonica*), which contains ursolic acid, prevented dexamethasone-induced muscle atrophy by inhibiting the muscle degradation pathway in Sprague–Dawley rats [196], whereas loquat leaf extract enhanced myogenic differentiation and muscle function in aged Sprague–Dawley rats [197] and healthy human adults [198].

A recent small clinical study revealed that an extract from *Schisandra chinensis* (SC) not only improves the thigh muscle strength of elderly women but also has a fatigue-improving effect [199]. SC fruit is a well-known traditional herb used for pharmacological purposes in Asian countries, such as Korea, China, and Japan. Studies in animals have suggested that SC extract has beneficial effects, such as decreasing protein degradation, increasing protein synthesis, and exhibiting antioxidant and anti-inflammatory effects on skeletal muscle fibers [200,201]. SC also improves mass, strength, and endurance in mice. Schisandrin A and schisandrin B are the major active ingredients in SC extract, which not only inhibit muscle atrophy in various animal experiments [200,201,202] but also promote muscle cell differentiation [203,204,205]. In addition to these, curcumin, resveratrol, catechin, soy protein, ginseng, and other substances known to have antioxidant, anti-aging, or anti-cancer properties are also being introduced as helpful in preventing sarcopenia without specific side effects on the human body [206]. However, the use of herbal supplements for the treatment and prevention of sarcopenia is not widely recommended until further research establishes their safety and efficacy in humans. 

## 11. Conclusions and Future Direction

Sarcopenia is an increasingly important health issue worldwide, with a prevalence of 5–13% among those aged 60–70 years and up to 50% among those over 80 years old [207,208]. As of 2017, the global population older than 60 years was estimated to be 962 million, which is projected to increase to 1.4 billion by 2030, with one in six people being aged 60 years or over. By 2050, this population is expected to double to 2.1 billion [209], and the number of individuals aged 80 years is predicted to triple to 426 million. Considering these statistics, sarcopenia, which currently affects over 80 million people, is conservatively expected to affect more than 320 million people over the next 30 years.

However, confirming the diagnosis of sarcopenia is challenging. Although comprehensive measurements used in research provide accurate results, they are often not feasible in clinical settings and may not affect patient care planning. Exercise remains the preferred method for managing sarcopenia; however, implementing an exercise program can be difficult for various reasons. The role of nutrition in the prevention and treatment of sarcopenia is not yet fully understood, and there is an ongoing debate regarding the optimal level of protein intake. However, it is generally recommended to ensure adequate protein intake and replace deficient nutrients and vitamins [210].

Future research should investigate the biological mechanisms that contribute to the development of sarcopenia and identify more accurate diagnostic biomarkers. Since 2016, sarcopenia has been recognized as a disease and assigned disease codes in numerous advanced countries worldwide, including the United States, Japan, and South Korea [72,124,211]. Multifaceted efforts are being made for the prevention and treatment of sarcopenia. Particularly, with the ongoing release of recent research findings on gene expression changes and single nucleotide polymorphisms associated with sarcopenia/frailty, as well as their associations with other diseases through genome-wide association studies and more [212,213], it is anticipated that in the near future, not only the approval of primary therapeutic agents but also the discovery of preventive measures will become possible.

## Figures and Tables

**Figure 1 biomedicines-11-01635-f001:**
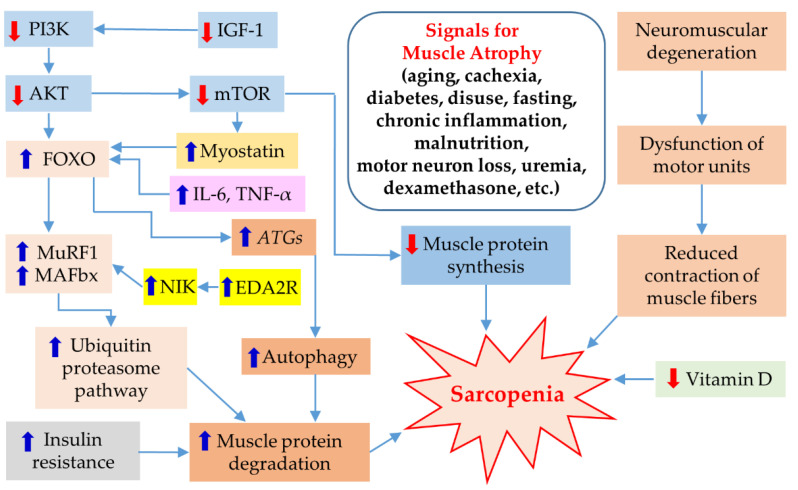
Schematic of the molecular mechanism of sarcopenia induced by several signals. AKT, protein kinase B; *ATGs*, autophagy genes; EDA2R, ectodysplasin A2 receptor; FOXO, forkhead box O; IGF-1, insulin-like growth factor-1; IL-6, interleukin-6; MAFbx, muscle atrophy F-box; mTOR, mammalian target of rapamycin; MuRF1, muscle RING Finger-1; NIK, NF-κB-inducing kinase; PI3K, phosphoinositol-3-kinase; TNF-α, tumor necrosis factor-α.

**Table 1 biomedicines-11-01635-t001:** Recent clinical status of SARMs.

Title	Intervention	Conditions	Primary Outcome	Phase	Status	Clinical Trials Identifier	Refs.
GTx-024 as a treatment for stress urinary incontinence in women	GTx-024	Stress urinary incontinence	The mean percent change in number of stress incontinence episodes/day as assessed by patient completion of the 3-day voiding diary	Phase 2	Completed	NCT02658448	[109]
Study to assess enobosarm (GTx-024) in postmenopausal women with stress urinary incontinence (ASTRID)	GTx-024, placebo	Stress urinary incontinence	Number of participants with a ≥50% reduction from baseline in the mean number of stress incontinence episodes per day at week 12	Phase 2	Completed	NCT03241342	[110]
Durability extension study to assess clinical activity and safety of enobosarm (GTx-024) in stress urinary incontinence	GTx-024, matching placebo	Stress urinary incontinence	Durability of response, stress incontinence	Phase 2	Terminated	NCT03508648	[111]
Study of GTx-024 on muscle wasting (cachexia) cancer.	GTx-024, placebo	Cachexia	The efficacy of GTx-024 on total body lean mass	Phase 2	Completed	NCT00467844	[112,113]
Add-on study for protocol G200802 (NCT02463032): effect of GTx-024 on maximal neuromuscular function and lean body mass	GTx-024, 9 or 18 mg	ER+ and AR+ breast cancer	Maximal power production (assessed by inertial-load cycle ergometry)	Phase 2	Completed	NCT02746328	[114]
Effect of GTx-024 on muscle wasting in patients with NSCLC on first line platinum	GTx-024, placebo	Muscle wasting, NSCLC	Physical function (measure is the percentage of subjects at day 84 with stair climb power change), lean body mass (measure is the percentage of subjects at day 84 with lean body mass change)	Phase 3	Completed	NCT01355497	[115,116]
Phase III study on the effect of GTx-024 on muscle wasting in patients with NSCLC	GTx-024, placebo	Muscle wasting, NSCLC	Physical function (measure is the percentage of subjects at day 84 with stair climb power change), lean body mass (measure is the percentage of subjects at day 84 with lean body mass change)	Phase 3	Completed	NCT01355484	[116,117]
Enobosarm and anastrozole in pre-menopausal women with high mammographic breast density	Enobosarm (GTx-024)	Mammographic density	Mammographic breast density, breast tissue elasticity	Phase 1	Completed	NCT03264651	[118]
Study to evaluate the safety and efficacy of 13 weeks of the SARM GSK2881078 in chronic obstructive pulmonary disease (COPD)	GSK2881078, matching placebo	Cachexia	Change from baseline in SBP, DBP, heart rate and urinalysis parameter, number of participants with SAEs, percentage change from baseline in maximum leg press strength following 1 RM	Phase 2	Completed	NCT03359473	[119,120]

AR, androgen receptor; DBP, diastolic blood pressure; ER, estrogen receptor; NSCLC, non-small-cell lung cancer; SAEs, serious adverse events; SARM, selective androgen receptor modulator; SBP, systolic blood pressure; 1 RM, 1 repetition maximum.

**Table 2 biomedicines-11-01635-t002:** Recent clinical status of myostatin inhibitor drugs.

Title	Intervention	Conditions	Primary Outcome	Phase	Status	Clinical Trials Identifier	Refs.
A phase 2 study to evaluate the safety, efficacy, pharmacokinetics, and pharmacodynamics of PF-06252616 in Duchenne muscular dystrophy	PF-06252616, placebo	Duchenne muscular dystrophy	Number of participants with TEAEs by week 49, change from baseline on the 4SC as compared to placebo at weeks 17, 33, and 49	Phase 2	Terminated	NCT02310763	[126,127,128,129,130,131,132,133]
An open-label extension study to evaluate safety of PF-06252616 in boys with Duchenne muscular dystrophy	PF-06252616	Duchenne muscular dystrophy	Number of participants with dose reduced or temporary discontinuation due to AEs, number of participants with severe TEAEs	Phase 2	Terminated	NCT02907619	[127,134]
Efficacy and safety of apitegromab in patients with later-onset spinal muscular atrophy treated with nusinersen or risdiplam (SAPPHIRE)	Apitegromab, placebo	SMA, muscular atrophy	Main efficacy population: change from baseline in HFMSE total score	Phase 3	Recruiting	NCT05156320	[135]
Study evaluating MYO-029 in adult muscular dystrophy	MYO-029	Becker muscular dystrophy, facioscapulohumeral muscular dystrophy, limb–girdle muscular dystrophy	Safety assessment	Phase 1, Phase 2	Completed	NCT00104078	[136]
An active treatment study of SRK-015 in patients with type 2 or type 3 spinal muscular atrophy (TOPAZ)	SRK-015	SMA	Change from baseline in the RHS total score at day 364, change from baseline in HFMSE total score at day 364	Phase 2	Active, not recruiting	NCT03921528	[137]
Long-term safety and efficacy of apitegromab in patients with SMA who completed previous trials of apitegromab-ONYX (ONYX)	Apitegromab	SMA	Evaluate the long-term safety and tolerability of apitegromab in patients with type 2 and type 3 SMA	Phase 3	Not yet recruiting	NCT05626855	[138]
Study of efficacy and safety of bimagrumab in patients after hip fracture surgery	Bimagrumab	Muscle wasting (Atrophy) after hip fracture surgery	Change from baseline in total lean body mass measured by DXA at weeks 12 and 24	Phase 2	Completed	NCT02152761	[139,140]
Safety, pharmacokinetics, and efficacy of bimagrumab in overweight and obese patients with type 2 diabetes	BYM338 10 mg/kg, placebo	Diabetes mellitus type 2	Change from baseline in total body fat mass by DXA at week 48	Phase 2	Completed	NCT03005288	[141,142]
An extension study of the efficacy, safety, and tolerability of BYM338 (Bimagrumab) in patients with sporadic inclusion body myositis who previously participated in the core study CBYM338B2203	Bimagrumab, placebo	sIBM	Number of participants with AEs, SAEs, and deaths, change from core study baseline in 6MWD	Phase 3	Completed	NCT02573467	[143,144]
Dose range finding study of bimagrumab in sarcopenia	Bimagrumab, placebo	Sarcopenia	Change from baseline in Total SPPB score to week 25	Phase 2	Completed	NCT02333331	[145,146]
Study of long-term safety, efficacy tolerability of BYM338 in patients with sIBM (BYM338)	BYM338 (Bimagrumab)	sIBM	Number of participants with AE a measure of safety and tolerability	Phase 2 Phase 3	Completed	NCT02250443	[147,148]
Efficacy and safety of bimagrumab/BYM338 at 52 weeks on physical function, muscle strength, mobility in sIBM patients (RESILIENT)	BYM338/ Bimagrumab, placebo	sIBM	Change from baseline in 6MWD test at week 52	Phase 2 Phase 3	Completed	NCT01925209	[149,150]
A multi-center study to assess the effects of BYM338 on skeletal muscle in sarcopenic adults	BYM338, placebo	Skeletal muscle	Muscle volume of the thigh (measurement gathered using MRI, magnetic resonance imaging)	Phase 2	Completed	NCT01601600	[151]
Efficacy, safety, and tolerability of BYM338 in patients with sIBM	BYM338, placebo	sIBM	Effect of BYM338 on thigh muscle volume by MRI	Phase 2	Completed	NCT01423110	[152,153]
A 24-week off-drug extension study in sarcopenic elderly who completed treatment in the 6-month core study	Bimagrumab, placebo	Sarcopenia	SPPB total score at week 49	Phase 2	Completed	NCT02468674	[154]
BYM338 in COPD patients with cachexia	BYM338, placebo	COPD with cachexia	Percentage change from baseline of TMV by MRI Scan at week 4, 8, 16, and 24	Phase 2	Completed	NCT01669174	[155,156]
Clinical study of BYM338 for the treatment of unintentional weight loss in patients with cancer of the lung or the pancreas	BYM338 active drug, placebo	Cachexia	Percentage change from baseline of TMV by MRI Scan at week 8	Phase 2	Completed	NCT01433263	[157]
A study of LY2495655 in older participants undergoing elective total hip replacement	LY2495655, placebo	Muscular atrophy	Change from baseline in aLBM at week 12	Phase 2	Completed	NCT01369511	[158,159]
A study in older participants who have fallen and have muscle weakness	LY2495655, placebo	Muscle weakness	Change from baseline to 24 weeks endpoint in aLBM	Phase 2	Completed	NCT01604408	[160,161]
Study of the safety and efficacy of REGN1033 (SAR391786) in patients with sarcopenia	REGN1033 (SAR391786), placebo	Sarcopenia	Percent change in total lean body mass	Phase 2	Completed	NCT01963598	[162]
A study to evaluate the efficacy and safety of taldefgrobep Alfa in participants with spinal muscular atrophy (RESILIENT)	Taldefgrobep alfa, placebo	SMA, neuromuscular diseases	Efficacy of taldefgrobep alfa compared to placebo in change in the MFM-32 total score	Phase 3	Recruiting	NCT05337553	[163]
Clinical intramuscular gene transfer of rAAV1.CMV.huFollistatin344 trial to patients with Duchenne muscular dystrophy	rAAV1.CMV.huFollistin344	Duchenne muscular dystrophy	Number of DLT adverse events as assessed by 21 CFR 312.32.	Phase 1 Phase 2	Completed	NCT02354781	[164]
Follistatin gene transfer to patients with Becker muscular dystrophy and sporadic inclusion body myositis	rAAV1.CMV.huFollistatin344	Becker muscular dystrophy, sIBM	Safety	Phase 1	Completed	NCT01519349	[165,166]

AE, adverse event; aLBM, appendicular lean body mass; COPD, chronic obstructive pulmonary disease; DLT, dose limiting toxicity; DXA, dual energy X-ray absorptiometry; HFMSE, hammersmith functional motor scale expanded; MRI, magnetic resonance imaging; MFM-32, 32 item motor function measure; RHS, revised hammersmith scale; SAEs, serious adverse events; sIBM, sporadic inclusion body myositis; SMA, spinal muscular atrophy; SPPB, short physical performance battery; TEAEs, treatment-emergent adverse events; TMV, thigh muscle volume; 4SC, 4 stair climb; 6MWD, 6 min walking distance test.

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
