# Peer review of "Pathogenesis, Intervention, and Current Status of Drug Development for Sarcopenia: A Review"

_biomedicines, 2023, doi:10.3390/biomedicines11061635_

Round 1
Reviewer 1 Report
The present article “Pathogenesis, Intervention, and Current Status of Drug Development for Sarcopenia: A Review” is interesting and well-written. However, the authors should address the following minor concern.
· Authors are suggested to include the reference in the last paragraph of Page 2; 3.2 sub-heading “Research has revealed that both IL-6 and TNF-α levels are elevated in the skeletal muscles of older individuals.”
Author Response
- Point 1: Authors are suggested to include the reference in the last paragraph of Page 2; 3.2 sub-heading “Research has revealed that both IL-6 and TNF-α levels are elevated in the skeletal muscles of older individuals.”
- Response 1: Thank you for your comments. In response to your comments, we have added References number 28 and 29 to the last paragraph on page 3 of the manuscript.
We tried our best to improve the manuscript and made some changes in the manuscript. These changes will not influence the content and framework of the paper. And here we did not list the changes but marked them in red in the revised paper.
We appreciate for Editors/Reviewers’ warm work earnestly and hope that the correction will meet with approval.
Once again, thank you very much for your comments and suggestions.

Reviewer 2 Report
This is a well-written paper focusing on pharmacological and non-pharmacological methods to prevent and treat sarcopenia.
I have several suggestions.
Figure 1. Perhaps, mentioning EDA2R-NIK pathway near MuRF1/MAFbx will add more novelty to the scheme (PMID: 37165186).
One of the aims of the review was to describe molecular mechanisms of sarcopenia. However, this part was not written in detail including recent findings in the field. Based on this, new therapeutic strategies can be developed. For example, the use of exercise mimetics like GW501516 (PMID: 33397034) can activate PPARD gene which is linked with sarcopenia. I therefore, recommend to mention more sarcopenia / frailty related genes in the review (for example, PMID: 36771461 and 36928559 etc.), and to write about potential use of exercise / diet / drugs to modulate their expression for the prevention and treatment of sarcopenia.
Author Response
- Point 1: Figure 1. Perhaps, mentioning EDA2R-NIK pathway near MuRF1/MAFbx will add more novelty to the scheme (PMID: 37165186).
- Response 1: Thanks for the suggestion. We have included the EDA2R-NIK pathway in Figure 1 of the manuscript and cited the reference (PMID: 37165186) as Reference number 59.
- Point 2: One of the aims of the review was to describe molecular mechanisms of sarcopenia. However, this part was not written in detail including recent findings in the field. Based on this, new therapeutic strategies can be developed. For example, the use of exercise mimetics like GW501516 (PMID: 33397034) can activate PPARD gene which is linked with sarcopenia. I therefore, recommend to mention more sarcopenia / frailty related genes in the review (for example, PMID: 36771461 and 36928559 etc.), and to write about potential use of exercise / diet / drugs to modulate their expression for the prevention and treatment of sarcopenia.
- Response 2: Thanks for your comments. We have revised the manuscript by incorporating additional descriptions related to exercise mimetics, including GW501516 (page 17; ‘9. Exercise Mimetics’). In addition, we have revised the conclusion by adding references (PMID 3661461; PMID 36928559) related to genes associated with sarcopenia/frailty.
We tried our best to improve the manuscript and made some changes in the manuscript. These changes will not influence the content and framework of the paper. And here we did not list the changes but marked them in red in the revised paper.
We appreciate for Editors/Reviewers’ warm work earnestly and hope that the correction will meet with approval.
Once again, thank you very much for your comments and suggestions.

Reviewer 3 Report
This is a very interesting narrative review on the current situation about sarcopenia pharmacological treatments.
I can give a good evaluation to this paper but I have several minor comments.
1) please into the abstract define what is the purpose of this review
2) epidemiology part should be focused on different regions. Please create an heap map
3) since table 2 is based on myostatin inhibitors drugs, please create a figure highlighing the mechanisms of action
4) into the introduction please define the different phenotypes of sarcopenia, like sarcopenic obesity and osteosarcopenic visceral obesity, citing the article listed here Perna, S., Spadaccini, D., Nichetti, M., Avanzato, I., Faliva, M.A. and Rondanelli, M., 2018. Osteosarcopenic visceral obesity and osteosarcopenic subcutaneous obesity, two new phenotypes of sarcopenia: prevalence, metabolic profile, and risk factors. Journal of aging research, 2018.
4) since you treated the vitamin D, please define the possible action of a multingredient with leucine as primary therapy or supplementation supporting the citing the paper Rondanelli, M., Cereda, E., Klersy, C., Faliva, M.A., Peroni, G., Nichetti, M., Gasparri, C., Iannello, G., Spadaccini, D., Infantino, V. and Caccialanza, R., 2020. Improving rehabilitation in sarcopenia: a randomized‐controlled trial utilizing a muscle‐targeted food for special medical purposes. Journal of cachexia, sarcopenia and muscle, 11(6), pp.1535-1547.
the quality is acceptable
Author Response
- Point 1: Please into the abstract define what is the purpose of this review.
- Response 1: Thank you. We have already stated the purpose of the review in the abstract (lines 20-22). “In this review, we aimed to examine the latest information on the epidemiology, molecular mechanisms, interventions, and possible treatments with new drugs for sarcopenia.”
- Point 2: Epidemiology part should be focused on different regions. Please create an heap map.
Response 2: Thank you for your comments. Our review examines the latest information on epidemiology, molecular mechanisms, interventions, and possible treatments with new drugs for sarcopenia. Currently, there is a limited amount of research on global regional and racial differences in the prevalence of sarcopenia. Therefore, it would be challenging not only to conduct studies on the global regional epidemiology of sarcopenia but also to create a heatmap based on the findings. Moreover, this would be a significant project that would require separate research. We appreciate your understanding regarding this matter.
- Point 3: Since Table 2 is based on myostatin inhibitors drugs, please create a figure highlighing the mechanisms of action
- Response 3: Thanks for your suggestion. The molecular mechanisms of myostatin inhibitor drugs are well reported in the following papers, in addition to the references cited in our review paper [121-125].
- Rybalka, E.; Timpani, C.A.; Debruin, D.A.; Bagaric, R.M.; Campelj, D.G.; Hayes, A. The Failed Clinical Story of Myostatin Inhibitors against Duchenne Muscular Dystrophy: Exploring the Biology behind the Battle. Cells 2020, 9, 2657, doi:10.3390/cells9122657.
- Abati, E.; Manini, A.; Comi, G.P.; Corti, S. Inhibition of myostatin and related signaling pathways for the treatment of muscle atrophy in motor neuron diseases. Cell. Mol. Life Sci. 2022, 79, 374, doi:10.1007/s00018-022-04408-w.
- Liu, J.; Pan, M.; Huang, D.; Guo, Y.; Yang, M.; Zhang, W.; Mai, K. Myostatin-1 Inhibits Cell Proliferation by Inhibiting the mTOR Signal Pathway and MRFs, and Activating the Ubiquitin-Proteasomal System in Skeletal Muscle Cells of Japanese Flounder Paralichthys olivaceus. Cells 2020, 9, 2376, doi:10.3390/cells9112376.
- Han, H.Q.; Zhou, X.; Mitch, W.E.; Goldberg, A.L. Myostatin/activin pathway antagonism: molecular basis and therapeutic potential. Int. J. Biochem. Cell Biol. 2013, 45, 2333-2347, doi:10.1016/j.biocel.2013.05.019.
Table 2 aims to convey the clinical status of myostatin inhibitors drugs. Therefore, we will not include the detailed molecular mechanisms of myostatin inhibitors. We appreciate your understanding regarding this matter.
- Point 4: Into the introduction please define the different phenotypes of sarcopenia, like sarcopenic obesity and osteosarcopenic visceral obesity, citing the article listed here Perna, S., Spadaccini, D., Nichetti, M., Avanzato, I., Faliva, M.A. and Rondanelli, M., 2018. Osteosarcopenic visceral obesity and osteosarcopenic subcutaneous obesity, two new phenotypes of sarcopenia: prevalence, metabolic profile, and risk factors. Journal of aging research, 2018.
- Response 4: Thanks for your suggestion. We defined various phenotypes of sarcopenia in the introduction of the manuscript (line 44) and added the article as Reference number 7.
- Point 4: Since you treated the vitamin D, please define the possible action of a multingredient with leucine as primary therapy or supplementation supporting the citing the paper Rondanelli, M., Cereda, E., Klersy, C., Faliva, M.A., Peroni, G., Nichetti, M., Gasparri, C., Iannello, G., Spadaccini, D., Infantino, V. and Caccialanza, R., 2020. Improving rehabilitation in sarcopenia: a randomized‐controlled trial utilizing a muscle‐targeted food for special medical purposes. Journal of cachexia, sarcopenia and muscle, 11(6), pp.1535-1547.
- Response 4: Thanks for your comments. We added the article as Reference number 170.
We tried our best to improve the manuscript and made some changes in the manuscript. These changes will not influence the content and framework of the paper. And here we did not list the changes but marked them in red in the revised paper.
We appreciate for Editors/Reviewers’ warm work earnestly and hope that the correction will meet with approval.
Once again, thank you very much for your comments and suggestions.

Round 2
Reviewer 2 Report
Well done, I have no additional comments.